# Characterization of Urine Stem Cell-Derived Extracellular Vesicles Reveals B Cell Stimulating Cargo

**DOI:** 10.3390/ijms22010459

**Published:** 2021-01-05

**Authors:** Asmaa A. Zidan, Mohammed Al-Hawwas, Griffith B. Perkins, Ghada M. Mourad, Catherine J. M. Stapledon, Larisa Bobrovskaya, Xin-Fu Zhou, Plinio R. Hurtado

**Affiliations:** 1Health and Biomedical Innovation, Clinical and Health Sciences, University of South Australia, Adelaide, SA 5000, Australia; Zidaa001@mymail.unisa.edu.au (A.A.Z.); Mohammed.Al-hawwas@unisa.edu.au (M.A.-H.); Larisa.Bobrovskaya@unisa.edu.au (L.B.); 2Department of Medical Histology and Cell Biology, Faculty of Medicine, Alexandria University, Alexandria 21568, Egypt; Ghada.Mourad@alexmed.edu.eg; 3Centre of Excellence for Research in Regenerative Medicine Applications (CERRMA), Faculty of Medicine, Alexandria University, Alexandria 21568, Egypt; 4Department of Molecular and Cellular Biology, School of Biological Sciences, University of Adelaide, Adelaide, SA 5000, Australia; griffith.perkins@adelaide.edu.au; 5Centre for Orthopaedic and Trauma Research, University of Adelaide, Adelaide, SA 5000, Australia; Catherine.Stapledon@adelaide.edu.au; 6Department of Renal Medicine, Royal Adelaide Hospital, Adelaide, SA 5000, Australia; 7School of Medicine, University of Adelaide, Adelaide, SA 5000, Australia

**Keywords:** exosomes, B cells, BAFF, CD40L, mesenchymal stem cells, urine stem cells, extracellular vesicles

## Abstract

Elucidation of the biological functions of extracellular vesicles (EVs) and their potential roles in physiological and pathological processes is an expanding field of research. In this study, we characterized USC–derived EVs and studied their capacity to modulate the human immune response in vitro. We found that the USC–derived EVs are a heterogeneous population, ranging in size from that of micro–vesicles (150 nm–1 μm) down to that of exosomes (60–150 nm). Regarding their immunomodulatory functions, we found that upon isolation, the EVs (60–150 nm) induced B cell proliferation and IgM antibody secretion. Analysis of the EV contents unexpectedly revealed the presence of BAFF, APRIL, IL–6, and CD40L, all known to play a central role in B cell stimulation, differentiation, and humoral immunity. In regard to their effect on T cell functions, they resembled the function of mesenchymal stem cell (MSC)–derived EVs previously described, suppressing T cell response to activation. The finding that USC–derived EVs transport a potent bioactive cargo opens the door to a novel therapeutic avenue for boosting B cell responses in immunodeficiency or cancer.

## 1. Introduction

Extracellular vesicles (EVs) were first described in human plasma by Peter Wolf as a form of platelet secretion and were also known as platelet dust [1]. However, subsequent studies showed that nano–and micro–sized membrane–bounded vesicles are constitutively secreted by a variety of cells, including mesenchymal stem cells (MSCs) [2,3]. These vesicles are generated in two main ways. One way is by the inward budding of the late endosome membrane forming intraluminal vesicles of (40–150 nm), creating multivesicular bodies (MVB), which subsequently fuse with the cellular membrane, secreting these vesicles, exosomes. The second is by outward membrane budding, enclosing cytoplasmic content forming vesicles 100–1000 nm in size, known as ectosomes or micro–vesicles [4], and collectively known as EVs [5]. Structurally, EVs are lipid bilayer–bounded vesicles, which bear surface membrane proteins, and contain bioactive molecules such as microRNA, mRNA, cytokines, growth factors, and soluble proteins, and thus serve as messengers from the cell of origin [6]. It has been shown that EVs mediate cell-cell communication and regulate important physiological processes such as the immune response and endocrine functions [7,8,9]. Recently, much attention has been directed at the immunomodulatory properties of MSC–derived vesicles, following the observation that MSCs exert anti–inflammatory and immuno–suppressive effects via direct cell-cell contact, as well as by cytokine and chemokine release, both in soluble form and loaded in EVs [10,11].

The anti–inflammatory effect of MSC–derived EVs has been shown to be useful in treating autoimmune and inflammatory conditions in several animal models, including graft–versus–host–disease (GVHD) [12], type 1 diabetes mellitus [13], asthma [14], and kidney inflammation [15], among other inflammatory conditions [16,17] {Galieva, 2019 #37}. The effect of these vesicles has been attributed to their cargo of cytokines like transforming growth factor–beta (TGFβ1), interleukin–6 (IL–6), interleukin 10 (IL–10), and hepatocyte growth factor (HGF), all of which are known to modulate immune responses [2]. The effect of the vesicles is also due to the presence of micro–RNAs (miRNAs), which modulate protein translation [18].

The discovery of urine stem cells (USCs), which originate from the kidneys [19], has simplified the isolation and culture of MSCs, offering a promising source of autologous stem cells for clinical use [20,21,22,23,24,25]. Similar to other types of MSCs, USCs secrete a wide range of biologically active molecules loaded into EVs, such as TGFβ1, vascular endothelial growth factor, angiogenin [26], DMBT1 [27], HGF, and IGF1 [28]. USC–derived EVs have also been proven effective at wound healing, ischemic limb repair, and prevention of diabetic renal complications, among other functions [26,28,29]. However, the possible role of USC–derived EVs in immune regulation is yet to be studied. Moreover, the cytochemical and ultra–structural criteria of the USCs, which could reflect their vesicle content in return, have not been addressed before.

In this paper, we report our findings on the characteristics of USC–derived EVs at the ultrastructural level, as well as their immune–modulatory effects on human peripheral blood mononuclear cells (PBMCs). We confirmed that USC–derived EVs are a heterogeneous population in size and density and that isolated EVs between 60 to 150 nm exhibit a typical cup–shape under the electron microscope. Isolated USC–derived EVs were tested for their effect on different immune cell populations. Similar to other MSC–derived EVs, USC–derived EVs exerted a suppressive effect on activated T cells. By contrast, USC–derived EVs had an unexpected, yet significant, stimulatory effect on B cell functions, inducing activation, proliferation, and IgM production. Surprisingly, in the lysate of USC–derived EVs, we found a range of molecules known to be potent B cell growth and survival factors, including IL–6, BAFF, APRIL, and CD40L.

These findings suggest that USCs could play a role in modulating local immune cells and, consequently, the renal immune environment. Our findings also suggest that USC–derived EVs may be of therapeutic value in conditions where boosting the immune response is desired, such as immunodeficiency or cancer immunotherapy.

## 2. Results

### 2.1. USC Isolation and Morphological Characterization

Following cell isolation, we sought to properly characterize the USCs at the phenotypic and ultrastructural levels to gain a better understanding of their possible EV content. Accordingly, cell clones were imaged between day seven and day ten of culture, where cells showed a stellate shape with multiple projections extending to nearby cells (Figure 1A). For further analysis of cell structure, USCs were stained with hematoxylin and eosin, followed by transmission electron microscopy (TEM) analysis.

**Cytochemical staining of the USCs:** hematoxylin and eosin staining of the USCs revealed large central vesicular nuclei, indicating active cells, and basophilic cytoplasm with perinuclear eosinophilia, denoting the protein–secreting nature of the cells. Kissing cells were noted in the field, denoting active division of the cells (Figure 1B). To investigate the perinuclear eosinophilia, TEM, and periodic acid–Schiff (PAS) staining were employed. PAS staining revealed the peri–nuclear magenta color of the cytoplasm (Figure 1C), concurring with the alpha glycogen–like dense granules found in the TEM (Appendix A), confirming high glycogen content in the cells.

### 2.2. MSC Nature of the USCs

After the characterization of the cells, we proceeded to confirm the MSC nature of the cells by testing for the differentiation potential to chondrocytes, adipocytes, and osteoblasts, as well as the molecular phenotype.

**Differentiation capacity of USCs:** After 21 days, successful mesenchymal lineage induction was evaluated by specific stains for each lineage. Accordingly, Toluidine blue applied to the induced chondrocyte pellet revealed blue staining for the deposited extracellular matrix (proteoglycan) (Figure 1D). Oil Red O applied to the induced adipogenic cells showed intracellular red lipid vacuoles (Figure 1E). Alizarin Red applied to the induced osteoblasts showed bright orange staining of the mineralization nodules between osteoblasts (Figure 1F).

**Phenotypic analysis of the USCs:** At passage four (P4), three different cell lines revealed that > 95% of the cells express MSC markers CD73 and CD105. Negligible expression of hematopoietic markers CD45, CD14, CD34, and HLA–DR, presented by one sample (Figure 1G), confirmed the purity of the isolation, further confirming USCs’ MSC nature.

### 2.3. Ultrastructural Examination of the USCs

The peculiar shape and pattern of the hematoxylin and eosin staining of USCs led us to investigate their ultrastructure using TEM. TEM examination revealed active cells with open phase nuclei, a wide network of rough endoplasmic reticulum (rER), abundant mitochondria, lysosomes, and pools of electron–dense granules (mimic glycogen bodies). Surprisingly, the cell surface showed plicae and was studded with vesicles of different sizes, denoting possible secretory activity of the cells (Figure 2A). Furthermore, cytosolic multivesicular bodies loaded with intra–luminal vesicles (exosomes), ranging in size from 60–110 nm, were visualized (Figure 2B). Analysis of the extracellular located vesicles showed sizes ranging from 50–1000 nm. The vesicles were heterogeneous in density (Figure 2C–E). Occasional budding of micro–vesicles (100–1000 nm) from the cell surface was noted, enclosing dense granules similar to the cytoplasm (Appendix A). In some instances, USCs cell membranes were observed to be enclosing exosome–sized vesicles, forming a pit, which may refer to its endocytosis (Figure 2F).

### 2.4. Characterization of Isolated EVs from USCs

Having confirmed the MSC characteristics of USCs through TEM observation of the exosome size range inside the MVB, we proceeded to isolate and characterize EVs from the USC supernatant. The isolated vesicles were characterized using TEM, nanoparticle tracking analysis (NTA), and western blot (WB). Using negative TEM staining of the isolated vesicles, we observed multiple cup–shaped vesicles with a similar size range between 60–130 nm (*n* = 19) by wide and close field (Figure 3A,B). The size of the isolated EVs was further confirmed by NTA, showing 85–140 nm, with a concentration of up to 11 × 10^11^ vesicles/mL with an average of 10^9^ vesicle/10^6^ cell (Figure 3C,D). WB analysis of the isolated vesicles showed the presence of CD63 and CD81—known EV surface markers—and the cytoplasmic marker, TSG101. Vesicles were negative to cytochrome C compared to their parental USC lysates, providing evidence of the purity of the EVs, purified from cell debris contamination (Figure 3E). The presence of CD63 was further confirmed using immunogold staining (Figure 3F).

### 2.5. USC–Derived EVs Activate T Cells without Inducing Proliferation

Having characterized the EVs from the USC supernatant, we proceeded to study their immune–modulatory properties by adding the EVs at a final concentration of 10 µg/mL to PBMC. We then measured their effect on the proliferation and activation of B and T cell populations. The EVs did not cause significant changes to the proliferation of the T cells upon the addition of EVs to the resting cells (12.5 ± 6.7% vs. 11.9 ± 5.5%, *n* = 5) in spite of enhanced early activation marker, CD69, expression (86.6 ± 0.9 vs. 2011 ± 682 MFI, *n* = 5) (Appendix A).

### 2.6. USC–Derived EVs Stimulate B Cells

In contrast to the effect EVs had on T cells, we found that USC–derived EVs significantly increased the expression of the activation marker, CD69 (195 ± 35 vs. 4700 ± 2000 MFI) in B cells and induced B cell proliferation (5.2 ± 3.4 vs. 26.9 ± 5.4 %, *n* = 5), compared to non–treated cells (Figure 4A,B). The expression of the co-stimulatory molecule CD40 was increased as the result of EV addition as well (1154 ± 250 vs. 1946 ± 400 MFI, *n* = 5), represented by Appendix A. In addition, the stimulatory effect of USC–derived EVs further enhanced the proliferation induced by the presence of CpGB (CpG oligodeoxynucleotides type B), known to induce strong B cell stimulation (52 ± 10 vs. 79 ± 6.4 %) (Mean ± SEM, *n* = 5) (Figure 4C).

To further confirm the effect of EVs on B cells, we proceeded to repeat the experiments, but this time using purified B cells. This yielded similar results, as shown in Appendix A.

To study the effect of the vesicles on IgM antibody secretion, we measured the antibody concentration in the purified B cell supernatant. We found that the presence of USC–derived EVs significantly enhanced IgM antibody secretion (12,220 ± 1629 vs. 15,044 ± 1378 pg/mL) (Mean ± SEM, *n* = 6) (Figure 4H).

### 2.7. Uptake of USC–Derived EVs by B Cells

To confirm that the effect we had observed was indeed a result of the interaction of the vesicles with B cells, we measured the uptake of fluorescently labeled EVs by B cells using confocal microscope Z–stacking. Observation of isolated B cells incubated with fluorescent–labeled EVs for 24 h showed a physical association between EVs and B cells, both around and inside the B cells (Figure 4D–F). Using flow cytometry, we found that the fluorescence intensity of B cells was significantly higher following incubation with the fluorescence–labeled EVs, compared with B cells cultured alone (60 ± 10 vs. 2863 ± 325 MFI) (Mean ± SEM, *n* = 3). Furthermore, we found a correlation between B cell activation based on elevation in CD69 and increased level of fluorescence, suggesting an association between EV and B cell activation (Figure 4G). This correlation suggests the role of EVs as a B cell stimulant.

### 2.8. USC–Derived EVs Are Carriers for B Cell Stimulants

To elucidate possible mechanisms behind the B cell stimulation by USC–derived EVs, we collected and analyzed the supernatants for known B cell stimulants. We observed that the presence of EVs resulted in a significant increase in IL–6 (1600 ± 575 vs. 6476 ± 1160 pg/mL), IL–10 (7.5 ± 3.4 vs. 30.5 ± 7.9 pg/mL), TNFα (5.7 ± 1.3 vs. 38.6 ± 5.3 pg/mL), and soluble CD40L (40.7 ± 10.9 vs. 105.9 ± 10.4 pg/mL) concentration compared with PBMCs alone (Mean ± SEM, *n* = 3) (Figure 5A,B). This observation prompted us to assess if the source of these cytokines was the EVs. We found that 10 µg/mL USC–derived EVs contained IL–6 (3400 ± 250 pg/mL), BAFF (410 ± 20 pg/mL), APRIL (92 ± 2 pg/mL), CD40L (75 ± 8 pg/mL), INFγ (22 ± 2.5 pg/mL) and traces of IL–10 and TNFα (*n* = 3) (Figure 5C).

To further confirm the presence of these cytokines in the EVs, we assessed CD40L expression in the isolated EVs using a monoclonal antibody to CD40L and nanogold–conjugated (6 nm) antibodies. TEM showed the presence of the 6 nm gold particles bound to CD40L on the EVs (Figure 5D). Unfortunately, our examination did not permit us to determine the accurate localization of the CD40L inside the EVs.

The cytoplasmic staining of USCs with mAb to CD40L also confirmed the high expression of the molecule inside the cells (Figure 5E).

### 2.9. USC Express Receptor for the B Cell Stimulant, BAFF

In order to explore the possibility of EVs exerting an autocrine/paracrine effect on USCs, we probed for the presence of the receptor for BAFF (BAFFR) using confocal microscopy. Surprisingly, we found that USC had high BAFF receptor expression (Figure 5F), which we further confirmed using flow cytometry (Figure 5G).

### 2.10. USC–Derived EVs Effect on Activated T Cells

Given that several studies have reported the inhibitory effect of MSC–derived EVs on activated T cells [30,31,32,33], we evaluated the overall effect of USC–derived EVs on T cells in the presence of anti–CD3/CD28 bead stimulation, mimicking physiological T cell activation by antigen–presenting cells. As expected, the addition of anti–CD3/CD28 beads to PBMCs resulted in strong T cell proliferation, which was significantly suppressed by the addition of EVs (72.6 ± 7.3 vs. 49.8 ± 4.7%) (Mean ± SEM, *n* = 5) (Figure 6A), (Appendix A).

Inducing apoptosis of activated T cells by MSCs has been described as one of the mechanisms by which they mediate T cell immune suppression [34,35]. Therefore, we measured the EVs’ effect on T cell viability. We observed that EVs also caused a reduction in the percentage of viable T cells, only in the culture condition of anti–CD3/CD28 activation (99 ± 1.5 vs. 82 ± 5 %, *n* = 3), as shown by a representative sample (Figure 6B–D).

Of note, in the presence of the anti–CD3/CD28 T cell stimuli in the PBMC culture resulted in an increase in CD19^+^ B cell proliferation. However, EVs decreased B cell proliferation (92.6 ± 2.6 vs. 86.7 ± 1.5%) (Mean ± SEM, *n* = 4), as shown in Appendix A. This inhibition could correlate with the inhibitory effect of T cells by EVs mentioned above, given that B cell proliferation in this condition is T cell-dependent [36]. Although interesting, the study of effect of USC’s EVs on T cells was beyond the aim of this study of effect of USC’s EVs on T cells was beyond the aim of this study.

## 3. Discussion

Extensive in vitro and in vivo characterization of the biological properties of EVs has paved the way for their use as a novel cell-free therapeutic modality [37,38]. This is particularly true for MSC–derived EVs, due to their rich content of biologically active molecules such as miRNAs and cytokines, including IL–10, TGF–β, Indoleamine 2,3-dioxygenase (IDO), and PGE–2, all of which have anti–inflammatory functions [38,39]. Consequently, they have been successfully used to decrease neuroinflammation in neurodegenerative diseases and to suppress autoreactive T cells in the treatment of GVHD and other inflammatory diseases [17,40,41].

The ease and cost–effectiveness of USC isolation has attracted much attention as a suitable alternative source of autologous MSCs for stem cell therapeutic applications [42]; however, characterization lags behind that of other MSCs.

Our results have shown that USCs have the characteristics of protein–secreting cells based on the basophilic staining of the cytoplasm and a wide network of rER at the ultrastructural level (Figure 2). This suggests high protein synthesis with potential high protein content in their EVs, in line with previous studies [26,27,28]. Furthermore, the presence of pools of dense granules with abundant plica and secretory vesicles studded to the cell membrane confirmed their property as active secretory cells.

Higher magnification images showed outward budding of membranes, enclosing similar dense granules, thereby forming heterogeneous populations of micro–vesicles, up to 1000 nm. The appearance of some of these dense granules was assumed to be alpha glycogen granules. This was confirmed using PAS staining, showing the magenta color of the cells. The presence of high glycogen content in USCs might be of therapeutic advantage in an ischemic environment, as reported for MSCs [43]. This could be associated with the report of IGF–1 presence in USCs’ EVs [28], as IGF–1 is known to induce intracellular glycogen synthesis [44].

TEM analysis of USCs showed the presence of MVB containing vesicles of different sizes, ranging from 60 to 110 nm, in line with previous reports [45]. Having confirmed the exosome size range by TEM, we proceeded to isolate USC–derived EVs of a similar size range from the USC culture supernatant using differential centrifugation [5,46]. In alignment with MISEV guidelines, the EVs were characterized by three different methods: nanoparticle analysis, morphologically by TEM, and by expression of phenotypic markers. Contamination of our preparation with cell debris was excluded by the negative expression of the cellular marker, cytochrome C [47]. Nano–Sight analysis of the EV pellets revealed a homogenous population of EVs less than 150 nm in diameter. This was further confirmed by TEM, where we observed an EV size range of 60–130 nm, and the classical cup–shaped morphology described with a similar staining method [48].

The size range and electron microscopy (EM) morphology of the isolated vesicles excluded apoptotic bodies. Moreover, Trypan blue staining of the cells before collecting the EVs showed >95% viability. Furthermore, an examination of the cells by TEM did not reveal any apoptotic features. Apoptotic bodies are larger–sized vesicles (500 nm to 4000 µm) that are released from the cells undergoing programmed cell death as a result of membrane disintegration and blebbing organelle remnants [49].

Regarding their immunomodulatory capacity, we found that the presence of USC–derived EVs had a significant stimulatory effect on B cells, as demonstrated by the increased expression of the activation marker, CD69, and the enhanced proliferation of resting B cells. This was without stimulating T cell proliferation, excluding mismatch or T cell-dependent stimulation, while synergizing with the stimulatory effect of unmethylated CpGB oligonucleotides (Figure 4).

B cell uptake of the EVs was confirmed by confocal microscope examination, which showed intracellular localization of USC–derived EVs, as well as their attachment to the surface of B cells after 24 hours’ incubation. Similar studies reported that EVs mediate their actions through different mechanisms: either by internalization or fusion with cell membranes delivering its content, including miRNAs, or attachment to surface receptors to trigger an intracellular signaling pathway [38,50,51,52]. A correlation between the increase of CD69 expression and the increase of B cells carboxy fluorescein succinimidyl ester (CFSE) fluorescence was observed when B cells were incubated with CFSE labeled EVs, denoting that physical association was mediating B cell activation.

In addition, EVs caused a significant increase in IgM, IL–6, IL–10, TNFα, and CD40L concentration in the supernatants, corroborating the profound effect of EVs on these cells. In order to understand this effect, we measured the presence of B cell-related cytokines in USC–derived EVs and found the presence of INFγ, IL–6, BAFF, APRIL, and CD40L, all known to be B cell stimulants. These play a role in B cell survival and cell cycle progression and are essential for an effective immune response [53,54,55].

Although CD40L can be found in a soluble form [56], CD40L is a type II transmembrane protein, and as such, it is mainly expressed in the cell membrane of activated T cells [57]. Here we showed that it is also expressed by USCs and is present in their EVs.

Contrary to USC–derived EVs, most studies of MSC–derived EVs on B cells have reported inhibitory effects [30,58]. However, the effect exerted by the MSCs themselves on B cells is still controversial between stimulation and inhibition [59,60,61,62,63], believed to be attributed to differences in B cell populations and culture conditions [64]. Analysis of these MSC–derived EVs showed the presence of similar miRNAs, IL–6, and IL–10; however, BAFF, APRIL, or CD40L have been reported in basal conditions. Nevertheless, priming the MSCs with TLR–4 has been shown to induce BAFF secretion [65].

Regarding T cells, we found that USC–derived EVs suppressed T cell proliferation under stimulatory conditions, in line with the previously reported immunosuppressive capacity of USCs and MSCs on T cells [34,66,67]. This effect has been extensively studied in MSCs and has been associated with the secretion of immunosuppressive cytokines [68,69] and miRNA, in particular miRNA146–5p, which was enriched in USC’s EVs, as reported previously [70,71]. However, further studies are warranted for full characterization of the effects of USC–derived EVs on T cells to address, for example, the EVs uptake, a more comprehensive study of the T cell activation, and the molecules responsible for their observed immune suppressive effects on these cells.

The different effects of the USC–derived vesicles on T and B cells, suppressing the former and stimulating the latter, is intriguing in terms of its biological function; however, the diversity of functions of the molecules they carry could support the finding. For example, IL–10 might be contributing to both effects, given its well know inhibitory effect on T cell functions [72,73,74] and stimulatory effects on B cells, contributing to their proliferation, differentiation, and antibody secretion [72]. Besides, the presence of TGFβ in these vesicles, previously reported [26], could be contributing to the inhibitory effect on T cell functions [75] whilst positively modulating B cell functions [76]. In addition, apart from the regulatory properties of the miRNA described above on T cells, miRNAs are also able to engage TLR7 [77,78,79], the main player in B cell activation [80,81]. Yet, further studies might elucidate other mechanisms behind our observation.

There is increasing evidence supporting a role for EVs in modulating the cellular microenvironment [82,83,84,85]. The unexpected presence of these molecules in USC–derived EVs raises new questions about the biological function, in health and in disease, of USCs in the kidneys, where they normally reside. Given the potent biological activity of these molecules on the immune response, and the central role of immune cells in most kidney pathologies such as in systemic lupus erythematosus (SLE), their possible role requires scrutiny. Interestingly, the presence of molecules such as BAFF and APRIL in urine has been correlated with the degree of lupus nephritis [86]. In addition, the expression of BAFF by renal tubular epithelium has been previously reported and is thought to support the survival of intra–renal long–lived plasma cells [87]. Given the renal origin of the USCs, their EVs could also play a role in supporting these plasma cells. Interestingly, our study showed the presence of BAFFR on the USCs, suggesting a possible autocrine/paracrine role of this factor on the USCs themselves; however, the effect of BAFF on USC is yet to be addressed.

Our study looked at a limited number of proteins based on the EVs’ effect on B cells; however, the presence of other molecules contributing to this effect cannot be ruled out. This could be resolved by a broader protein analysis method such as mass spectrometry–based proteome profiling of these vesicles of the USC derived EVs.

Due to the low immunogenicity of EVs, their long half–life, and high delivery efficiency, EVs are a strong candidate for cell therapy replacement [88,89]. The B cell-stimulating properties of USC–derived EVs that we reported here could add a new advantage for using these EVs as delivery vehicles for immune therapeutics. Furthermore, CD40L–enriched EVs have been reported to induce maturation of DCs, activation of T cells, and elicit strong anti–tumor effects [90]. A recent study found that the increase in B cells in the local cancer environment is correlated with a better response to checkpoint inhibitors and, subsequently, an improved prognosis [91]. These facts, together with the tissue tropism properties of EVs, indicate the possibility of using these EVs as adjuvants for cancer immunotherapy [92].

## 4. Materials and Methods

### 4.1. Isolation and Expansion of Human USCs

Following the previously described protocol [27], USCs were isolated from five healthy male donors between 24–50 years old with no history of chronic illness or urinary tract infection in the previous three months. Informed consent was obtained according to the code of research ethics adopted by the University of South Australia (UniSA) Human Ethics Committee (approval no. 35945). Fresh urine samples (150 mL) were centrifuged within 1 h of obtainment at 400 g at room temperature for 10 min. The obtained pellet was washed twice in 1% phosphate–buffered saline (PBS) (Gibco, Waltham, MA, USA) by centrifugation at the same speed. The pellet was cultured on gelatin–coated wells in primary isolation media. After 24 h, 50% of media change was performed. After a further 48 h, the media was gradually replaced with expansion media. Starting from the fourth day, the media was changed every other day with 50% expansion media. The cells were incubated in 5% CO_2_ at 37 °C. Once the cells reached 80% confluence, they were trypsinized (Lonza Basel, Switzerland) and sub-cultured again at a lower density of 3000 cells/cm^2^ [27].

The primary isolation media composition consisted of Dulbecco’s Modified Eagle Media (DMEM) /F–12, supplemented with the renal epithelial growth media (REGM), Single Quote kit (Lonza, Basel, Switzerland), 10% fetal bovine serum (FBS; Gibco, Waltham, MA, USA), and antibiotics (P/S) containing 100 U/mL of Penicillin and 100 µg/mL of Streptomycin (Gibco, Waltham, MA, USA). The expansion media composition consisted of mixed (1:1 ratio) DMEM/F–12 and renal epithelial basal media (RERBM), supplemented with REGM BulletKit, 10% FBS, P/S, 1% nonessential amino acid (NEAA; Gibco, Waltham, MA, USA), 5 ng/mL basal fibroblast growth factor, platelet–derived growth factor, and epidermal growth factor (PeproTech, Rocky Hill, NJ, USA).

Cells at passages three to four were collected for characterization after confirming more than 95% viability using Trypan Blue (Sigma–Aldrich, St. Louis, MO, USA).

### 4.2. USCs Characterization

#### 4.2.1. Cytochemical Staining

USCs at passage four were fixed using 4% formaldehyde for 20 min, followed by staining by either hematoxylin and eosin stain or 1% periodic acid–Schiff (PAS), countered by hematoxylin stain for characterization [93,94].

#### 4.2.2. Immune Fluorescence Staining

USCs at passage three were cultured overnight in a 24–well plate over covered glass at 2.6 × 104 cells/cm^2^. Cells were then washed twice before fixing with 4% paraformaldehyde for 30 min at room temperature, followed by permeabilization with 0.25 Triton × 100 for 10 min. Cells were washed three times for 5 min each, and non–specific binding was blocked by buffer containing bovine serum albumin (BSA) (10 mg/mL) and glycine (22.52 mg/mL) for 30 min at room temperature. Cells were incubated with mAb to CD40L, BAFFR, and isotype control (Appendix A), diluted in PBS containing BSA (10 µg/mL), overnight in humidified chamber at 4 °C. Cells were washed as described above and incubated with Goat anti–mouse conjugated to either Alexa 488 or 647 diluted in PBS–BSA according to manufacture instructions for 1 h at room temperature. Cells were washed four times as described and mounted using DAPI mounting media (Sigma-Aldrich, St. Louis, MO, USA) for confocal microscope visualization.

#### 4.2.3. Transmission Electron Microscopy (TEM)

10^5^ USCs at passage four were collected by trypsinization, fixed with EM fixative buffer (4% paraformaldehyde/1.25% glutaraldehyde in PBS) for TEM examination [95]. The pellet was post–fixed in 2% osmium tetroxide, followed by serial washes in ascending ethanol concentrations for dehydration. The pellet was infiltrated with resin and left in the oven for polymerization. Ultra–thin sections were obtained using a diamond knife, to be visualized by Tecnai G2 Spirit TEM (FEI, Hillsboro, OR, USA).

### 4.3. MSCs Nature of the USCs

#### 4.3.1. Phenotypical Characterization of USCs

10^5^ USCs were collected by trypsinization and stained for CD73, CD105, CD14, CD34, CD45, and HLA–DR antibodies for 30 min at 4 °C. Then, the cells were washed once and analyzed by flow cytometry (FACS Canto; BD, Franklin Lakes, NJ, USA) for the percentage of cells expressing the marker in comparison to monoclonal antibody–stained cells. Three different USC samples were used for each antibody. The full list of antibodies is provided in Appendix A.

#### 4.3.2. USCs Differentiation

To test the mesenchymal differentiation potential, USCs were differentiated into adipocytes and osteoblasts as described before [96]. After 21 days of incubation in the differentiation media, with media being changed every third day, the cells were fixed by 4% formaldehyde and stained for Oil Red O to detect adipocytes and Alizarin Red to detect osteogenic nodules. For chondrocytes differentiation, USCs were cultured in suspension in 15 mL tubes with chondrogenic media for 21 days with media change every other day [96]. The pellet was then fixed and sectioned for staining with Toluidine blue. The detailed composition of the differentiation media can be found in Appendix A.

### 4.4. USCs Extracellular Vesicles Isolation

EVs isolation was performed according to a previously described protocol [97]. In brief, USCs were washed twice with PBS in a T75 flask and cultured in serum–free expansion media. The conditioned media was collected after 48 h and run through successive centrifugation steps with increasing centrifugation forces and durations aiming to isolate EVs, based on their sizes as follows: centrifuge at 300× *g* for 10 min to precipitate cells, followed by the first centrifuge for the supernatant at 2000× *g* for 20 min at 4 °C to precipitate any dead cells. The supernatant was collected and centrifuged at 10,000× *g* for 40 min at 4 °C to precipitate cell debris. Furthermore, the supernatant was filtered through a 0.45 μm strainer (Corning, NY, USA). Finally, the EVs were precipitated by ultra–centrifugation (Beckman Coulter Type,70 Ti rotor, Brea, CA, USA) at 100,000× *g* for 90 min at 4 °C [98]. The formed pellet was washed once with PBS using the same speed to eliminate concomitant proteins from the EV pellet. The obtained EV pellet was resuspended in 100 µL PBS.

The absence of Mycoplasma and Acholeplasma contamination to the USC and the isolated vesicles was confirmed by PCR analysis of the supernatant following a previously described protocol [99].

### 4.5. EV Characterization

The morphology and size distribution of EVs were examined by TEM and nanoparticle tracking analysis (Nano–sight NS300, Malvern, UK), respectively. Each EVs pellet was prepared from 150 mL media collected from 16 × 10^6^ cells/mL incubated for 48 h with a one–week interval between each collection (*n* =19).

The protein content of the EV pellet lysate and the USCs lysate was quantified using the Micro BCA™ Protein Assay Kit (Thermo–Fisher, Waltham, MA, USA) according to the manufacturer’s instructions. Then the sample absorbance was read with the VICTOR^3^ Multilabel Plate Reader (PerkinElmer, Waltham, MA, USA) at a wavelength of 562 nm [100].

#### 4.5.1. Western Blot

The EV markers (including CD63, CD81, TSG101) and the cellular marker Cytochrome C were analyzed in the USC lysate and USC–derived EVs according to the previously described western blot protocol [101]. Briefly, EV and USCs pellets were lysed using a 5 × RIPA buffer containing 1% protease inhibitor, and the protein concentrations were estimated, as mentioned above. Then, 10 µg protein from each sample was loaded onto 10% SDS–PAGE gels and blotted onto 0.2μm nitrocellulose membranes. The membranes were incubated with blocking buffer (5% nonfat dried milk in TBST) for 30 min and incubated overnight with primary antibodies including Rabbit anti–CD81(1:500), Rabbit anti–Cytochrome C (1:1000), mouse mAb to CD63 (1:500), and mouse anti–TSG101 (1:1000) at 4 °C. The membranes were then incubated with the corresponding secondary antibodies (1:20,000, Donkey anti–Rabbit IgG for CD81 and Cytochrome C, Goat anti–Mouse IgG for CD63 and TSG101), at room temperature for one hour (Appendix A). The blots were read using the Odyssey^®^ CLx imaging system (LI–COR Biosciences, Lincoln, NE, USA) and band intensity quantified using Image Studio Lite (LI–COR Biosciences, Lincoln, NE, USA). Standard of known molecular weight was included (Precision Plus Protein™ Dual Color Standards, Bio–Rad, Hercules, CA, USA) in every run for determination of the protein band molecular weight. The markers for exosome characterization were selected according to MISEV guidelines [5].

#### 4.5.2. Nanoparticle Tracking Analysis (NTA)

EVs (30 µL) were diluted in filtered PBS at a ratio of 1:1000 and injected into the sample chamber of the nanoparticle analyzer (NanoSight 300, Malvern, UK). Six readings were taken for each sample. The camera level and detection threshold were adjusted manually for each experiment, as recommended by the manufacturer. Particle size, distribution, and concentration were analyzed using NTA analytical software (NanoSight NTA 3.2, Malvern, UK).

#### 4.5.3. Transmission Electron Microscopy (TEM)

The EV pellet was suspended in the EM fixative Buffer (1:10 dilution). Then, 5 µL of the solution was loaded on carbon–coated, glow discharged grids, and negatively stained with 2% uranyl acetate [102]. After three washes, the grid was left to dry, to be visualized by TEM. Further identification of the USCs’ vesicles was made by TEM, examining the USCs to visualize the intraluminal vesicles or the extracellular located vesicles. The size of the different vesicles was analyzed using Image J (NIH, Bethesda, MD, US).

#### 4.5.4. Immunogold Labelling

Five microliters of the EV pellet were loaded onto Formavar–coated copper grids. After two PBS washes to the grid (1 min each), the grid was treated with 0.05 M glycine for 10 min to quench free aldehyde groups. The grid was blocked using 1% BSA for 30 min and then incubated with the primary antibody, CD63 (1:50) or CD40L (1:50) for 1 h at room temperature. The grids were washed twice with PBS, followed by incubation with the secondary antibody (Aurion Goat anti–mouse IgG conjugated to 6 nm gold particles, 1:50) for 1 h at room temperature. Finally, the grid was washed as above before negative staining with 2% uranyl acetate [102]. Matching isotype antibody was used as a negative control.

#### 4.5.5. EVs Cytokine Content

To measure the cytokines in EVs, 10 µg/mL of the EVs was lysed with equal volume RIPA buffer and cytokines quantified using a B cell panel 13 multi–analyte complex (BioLegend, San Diego, CA, USA) following the manufacturer’s protocol.

### 4.6. PBMC Isolation

Fresh blood samples were collected from five healthy male donors between 24 and 50 years old after obtaining consent, according to the UniSA Human Ethics Committee (approval no. 35760). PBMC were isolated by Ficoll density gradient [103]. Cells in the interface layer were collected and washed three times with PBS. Finally, the cell pellet was resuspended at 10^6^ cells/mL in IMDM media (Gibco, Waltham, MA, USA), supplemented with 10% FBS, 2 mM L–glutamine, and 1% P/S, defined as complete media (CM). For B cell isolation, EasySep™ negative selection was used following the manufacturer’s protocol (STEMCELL, Vancouver, Canada) achieving a purity above 97%.

### 4.7. Co–Culture Studies

PBMC or B cells (2.5 × 10^5^/mL) suspended in CM were cultured alone or in the presence of 10 µg/mL of freshly isolated EVs, in 24 well plates at 37 °C and 5% CO_2,_ for five days, as previously described [30,104,105]. Additionally, experiments were performed in the presence of CpG–B ODN 2006 (phosphorothioate 5′–tcgtcgttttgtcgttttgtcgtt–3′) at 3.2 µg/mL final concentration (Geneworks, Victoria, Australia), or with anti–CD28/CD3 beads at 4:1 cells to beads ratio (Dynabeads, Thermofisher, Waltham, MA, USA), in order to provide B or T cell stimulation respectively [106,107]. The experiments were repeated at least twice.

### 4.8. EV Uptake Study

To study EV uptake by B cells, the EVs were labeled with ExoGlow™ Protein EV Labelling Kit (System Biosciences, Palo Alto, CA, USA). Then, the stained EVs were added to 2.5 × 10^5^ B lymphocytes and cultured in complete media at 37 °C and 5% CO_2_ for 24 h. The cells were then collected and washed twice, centrifuged 400× *g*, and either stained for CD19 and CD69 for analysis by flow cytometry or fixed with 2% formaldehyde for observation with a confocal microscope. The fixed cells were then cytospinned (Cytospin 4, Thermofisher, Waltham, MA, USA) at 5000× *g* for 10 min. The cell pellet was mounted using DAPI mounting media (Sigma-Aldrich, St. Louis, MO, USA) and visualized in a confocal microscope (Olympus FV3000 Confocal Microscope, Tokyo, Japan) using the Z–stack technique to elaborate on the location of the vesicles.

### 4.9. Proliferation Study

Lymphocyte proliferation was assessed after labeling with carboxyfluorescein succinimidyl ester (CFSE) (CellTrace™, Thermofisher, Waltham, MA, USA) following the manufacturer’s protocol. Labeled lymphocytes 2.5X10^5^ were co-cultured with EVs for five days. Then, cells were collected, and median fluorescence intensity (MFI) for CFSE fluorescence was analyzed by flow cytometry [108]. Accordingly, the percentage of the divided cells was calculated using FlowJo V10 (BD, Ashland, OR, USA).

### 4.10. Flow Cytometry

To identify lymphocyte activation, 2.5 × 10^5^ PBMC were cultured with the EVs for 24 h, as prescribed above. The cells were collected and stained with CD3, CD19, CD69, and CD40 antibodies for 30 min at 4 °C, followed by washing with PBS containing 1% BSA. They were then analyzed by flow cytometry (FACS Canto; BD, Franklin Lakes, NJ, USA). Lymphocytes were selected according to their forward and side scatter. The viable population was gated through staining with propidium iodide, as previously described [109]. Median fluorescence intensity for each marker was calculated by FlowJo software (BD, Ashland, OR, USA). A full list of used antibodies is provided in Appendix A.

### 4.11. Antibody Quantification and Cytokines Assay

Immunoglobulin M production in the B cell culture supernatant in response to conditioning with EVs was evaluated by multiplex flow cytometry–based enzyme–linked immunosorbent assay (BioLegend, San Diego, CA, USA) following the manufacturer’s protocol. In contrast, the PBMC supernatant in response to EV incubation, was analyzed for cytokine content. For this purpose, a bead–based multiplex assay panel, B cell panel 13 multi–analyte complex was used following the manufacturer’s instructions. The obtained cytokines data were analyzed with LEGENDplex software (BioLegend, San Diego, CA, USA).

### 4.12. Statistical Analysis

Descriptive statistics and statistical inference were performed using the GraphPad Prism (GraphPad, San Diego, CA, USA) software statistical analysis package. Assuming a normal distribution, using the Shapiro–Wilk normality test and the paired student *t*–test analysis was used to compare the mean marker expression. To normalize the distribution, the cytokine results underwent prior logarithmic transformation. The results were considered significant only if the *p*-value was <0.05. The experiments were carried using five different individual samples and repeated at least three times unless otherwise stated.

## 5. Conclusions

In summary, USCs were found to have high glycogen content and to secrete a wide size range of extracellular vesicles. Having investigated the immune effects of USC–derived vesicles in a size range of 60–150 nm, we confirmed that these vesicles could achieve the previously described effect of USCs on the suppression of activated T cells [66]. However, surprisingly, we found that these vesicles were vehicles for potent B cell stimulants, including IL–6, BAFF, APRIL, and CD40L, which could explain their stimulatory effect on B cells. Furthermore, USCs expressed BAFF R, which suggests a possible para/autocrine role of these EVs on the USCs themselves. This finding provides novel insight into the immunomodulatory abilities of USC–derived EVs, paving the way for further studies on their possible in vivo use to boost immune response.

## Figures and Tables

**Figure 1 ijms-22-00459-f001:**
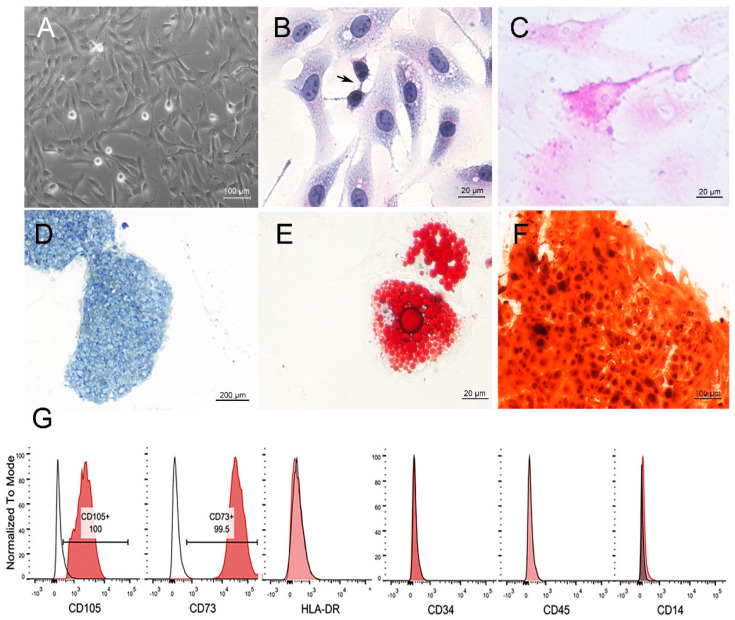
Urine stem cells (USCs) characterization; (**A**) Phase–contrast light micrograph (LM) at passage zero showing stellate shaped cells with membrane projection reaching nearby cells (original magnification ×100, scale bar = 100 µm). (**B**) hematoxylin and eosin staining of USCs at passage three, showing basophilic stellate cells with perinuclear eosinophilia, large central nuclei, and cytoplasmic projections Actively dividing cells were occasionally seen represented by kissing cells (arrow) (original magnification ×400, scale bar = 20 µm). (**C**) Periodic acid–Schiff (PAS) staining of USCs at passage three, showing perinuclear magenta color (original magnification ×400, scale bar = 20 µm). (**D**) Toluidine Blue staining of USC pellet following chondrogenic differentiation showing blue staining proteoglycan–rich extracellular matrix (original magnification ×40, scale bar = 200 µm). (**E**) Oil Red O staining of the induced adipogenic cells showing red staining of intracellular lipid (original magnification ×400, scale bar = 20 µm). (**F**) Alizarin Red staining of induced osteogenic cells showing a dark orange discoloration of the calcification nodules (original magnification ×100, scale bar = 100 µm). (**G**) A representative sample of immune phenotypical characterization of 10^5^ USCs (shaded curve) showing positive expression of CD105, CD73, and negative for HLA–DR, CD34, CD45, and CD14.

**Figure 2 ijms-22-00459-f002:**
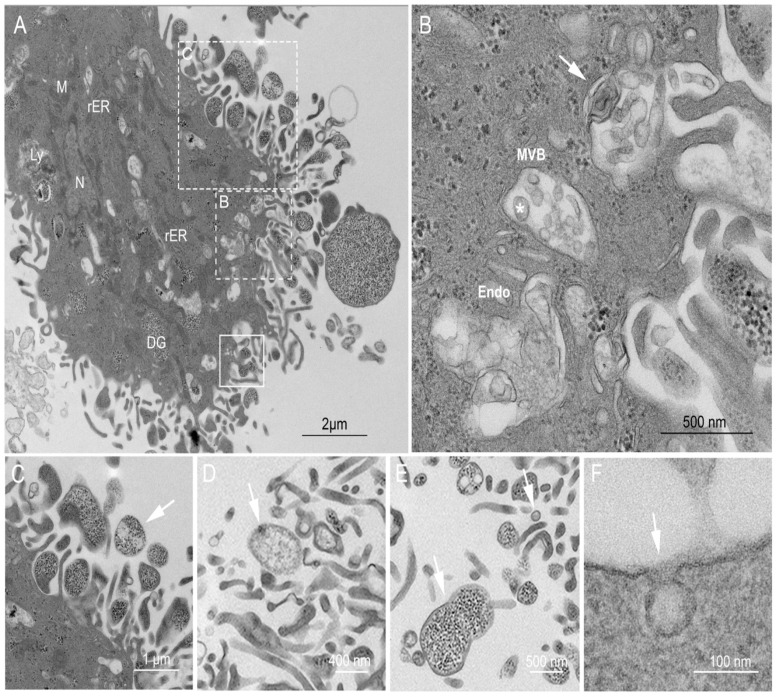
Transmission electron microscopy (TEM) characterization of USCs, and extracellular vesicles EVs morphology and Biogenesis; (**A**) USCs TEM where different cell structures can be visualized such as the nucleus (N), abundant mitochondria (M), rough endoplasmic reticulum (rER), lysosomes (ly), pools of dense granules (DG) and multivesicular body (MVB); magnified in (**B**), where biogenesis of exosomes can be observed in the three phases described, endosomes (Endo), MVB containing intraluminal vesicles of different sizes (Asterix) and fusing of MVB to the membrane with exosome release (Mic. Mag. ×13,000). The cell membrane exhibits multiple projections, membrane budding forming vesicles (solid square) and micro–vesicles (large dotted square, illustrated in (**C**) containing dense granules varying in size, shape, and content, (Mic. Mag. ×2900). (**C**–**E**) The pictures show the heterogenicity in sizes and density, varying dense granules containing larger vesicles between 150 nm–1µm, typical for micro–vesicles (**C**–**E**), and homogenous less granular vesicles < 150 nm, typical for exosomes (**E**,**F**), (Mic. Mag. ×2900). (**F**) Vesicle enclosed by USC membrane, mimic endocytosis (Mic. Mag. ×18,500) (Lead citrate/Uranyl acetate stain).

**Figure 3 ijms-22-00459-f003:**
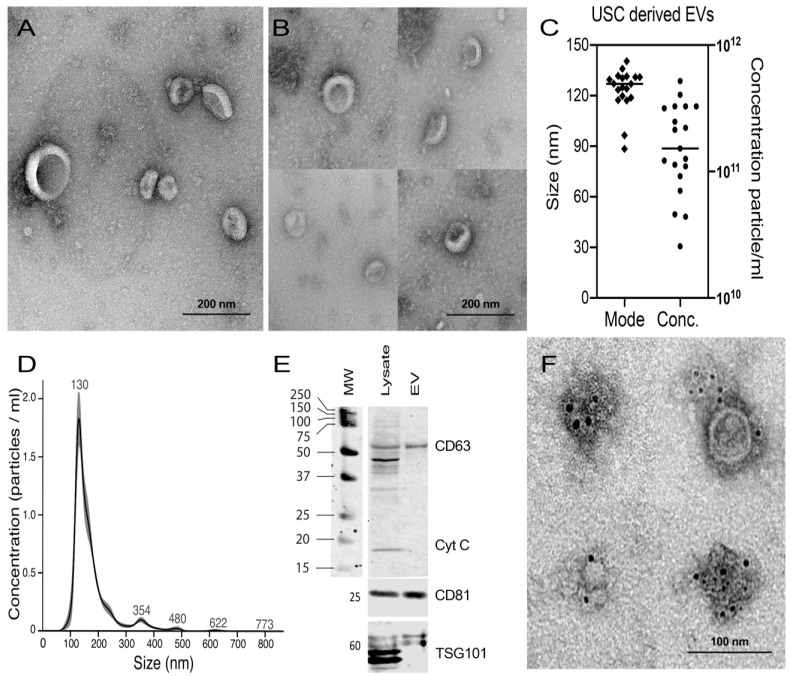
Characterization of USCs–derived EVs; (**A**,**B**) TEM negative staining of the isolated EVs showing cup–shaped vesicles with an average size of 110 nm visualized at lower magnification (**A**; Mic. Mag. ×23,000) and higher magnification (**B**; Mic. Mag. ×30,000) (2%Uranyl Acetate). (**C**) Histogram presentation of 19 EVs pellets collected from 1.6 × 10^7^ USCs analyzed for size and concentration, showing an average size of 122 nm and an average concentration of 1.90925 × 10^11^. (**D**) Size distribution curve of USCs isolated EVs particle concentration (×10^7^) vs. particle size mode, measured by nanoparticle tracking analysis (NTA) showing the average of the six technical replicate measurements for each exosome isolation by NanosightS300. (**E**) Western blot of USCs cell lysate and isolated the EVs for CD63, CD81, TSG101 antibodies as positive markers for EVs and Cytochrome C as mitochondrial membrane marker (cellular Marker) and negative marker for EVs. (**F**) Immune gold staining of the isolated vesicles for CD63 using 6 nm gold nanoparticles (2% Uranyl Acetate, scale bar = 100nm, Mic. Mag. ×30,000 (upper left) and ×23,000 rest).

**Figure 4 ijms-22-00459-f004:**
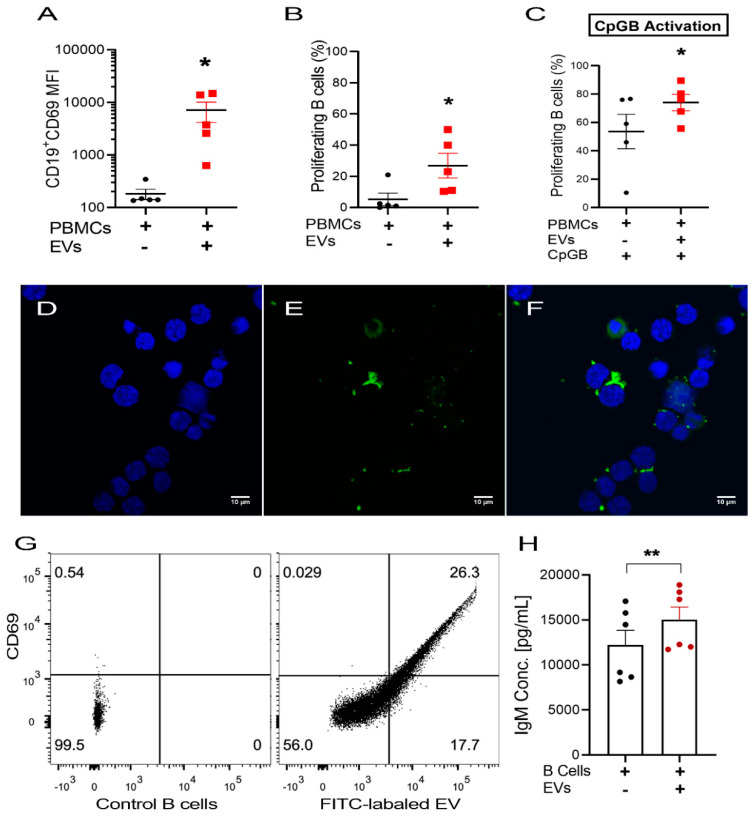
The effect of USCs EVs on B cells; (**A**) CD69 (early activation marker) expression on B cells population showing increased expression with EV co-culture. (**B**,**C**) Proliferation assay of the % proliferating B cells as the result of EVs co-culture in (**B**) resting and (**C**) CpGB–stimulated conditions showing significant enhancement of proliferation in both conditions in response to EV co-culture (*n* = 5). (**D**–**F**) Confocal microscopy images of DAPI (4′,6-diamidino-2-phenylindole) stained purified B cells (**D**) co-cultured with labeled EVs (**E**) for 24 h, (**F**) showing the presence of labeled EVs inside the cytoplasm and in aggregates attached to the cell surface (Mic. Magnification ×600, scale bar = 10 µm). (**G**) A representative sample of flow cytometry analysis of CD69^+^ B cell population for FITC labeled EVs uptake versus control. (**H**) Antibody analysis of the supernatant showing a significant increase of IgM with EV co-culture (*n* = 6). Values are presented as Mean ± SEM, *p*-values were determined by Paired Student’s *t*–tests. * *p* < 0.05, ** *p* < 0.01.

**Figure 5 ijms-22-00459-f005:**
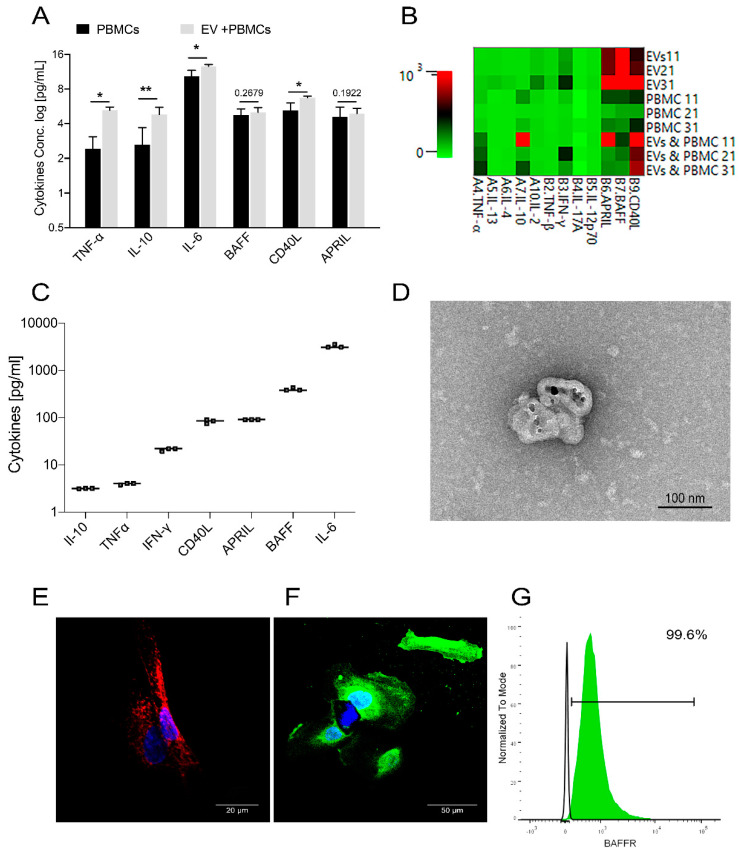
Cytokines Analysis of the EVs; (**A**) Cytokines analysis of EV co-culture supernatant run by Biolegend plex assay shows a significant increase in IL6, CD40L, IL10, and TNFα in response to EV co-culture (*n* = 3). (**B**) Heat map of the EV lysates, PBMCs, and the co-culture for cytokines concentration generated by Bio–legendplex software. (**C**) Cytokines analysis of 10 µg/mL EV lysates shows the expression of considerable amounts of IL–6, BAFF, APRIL, CD40L, and traces of IL–10, TNFα, and IFN–γ (*n* = 3). (**D**) Immune gold staining of the isolated vesicles for CD40L using 6 nm gold nanoparticles (Mic. Mag. ×23,000, 2% Uranyl Acetate). (**E**) Confocal microscope image of DAPI (blue) stained USC after permeabilization for CD40L mAb (red) (Mic. Mag. ×600, scale bar = 20 µm). (**F**) Confocal microscope image of DAPI (blue) stained USCs after staining with BAFFR mAb (green) (Mic. Mag. ×600, scale bar = 50 µm). (**G**) Flow cytometry analysis of 10^4^ USCs for BAFFR expression showing 99% positive cells (green shaded histogram) in relation to the isotype control (non–shaded histogram), representative sample. Values are presented as mean ± SEM and p–values were determined by paired Student’s *t*–tests. Cytokines data are prior to log treatment. * *p* < 0.05, ** *p* < 0.01.

**Figure 6 ijms-22-00459-f006:**
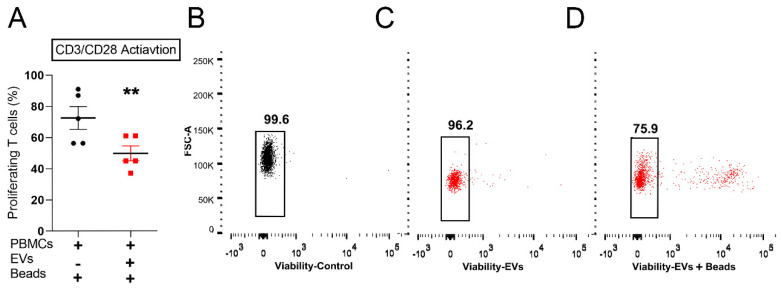
USCs EVs effect on T cells in response to anti–CD3/CD28 stimulation; (**A**) Effect of the EVs on the proliferation of T cells in the presence of anti–CD3/CD28–bead stimulation showing significant suppression of the T cell proliferation (represented by % proliferating cells, *n* = 5). Values are presented as mean ± SEM, and *p*-values were determined by paired *t*–test analysis. ** *p* < 0.01. (**B**–**D**) A representative sample of the viability assay of CD3^+^ (T cell) population where (**B**) control T cells viability, (**C**) viability in response to EVs co-culture. (**D**) Viability in response to EV co-culture in the presence of anti–CD3/CD28 beads. Note the decrease in the viable population in the activated T cells co-cultured with EVs.

## Data Availability

The data presented in this study are available on request from the corresponding author.

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
