# Peer review of "Characterization of Urine Stem Cell-Derived Extracellular Vesicles Reveals B Cell Stimulating Cargo"

_ijms, 2021, doi:10.3390/ijms22010459_

Round 1

Reviewer 1 Report

The authors sought to characterize the USC-derived EVs and studied their capacity to modulate the human immune response in vitro. I felt that a lot of effort was contributed to achieve this goal, and this manuscript is clear logic and well-written. However, it is mandatory to explain/correct the manuscript in some points.  

  1. How many biological replicates were performed in each analysis?
  2. What is the loading control in the Fig.3E?

Author Response

Dear Reviewer, 

We would like to thank you for reviewing our work and the constructive comments. We agree with your comments and have addressed them.

In this regards we have:

1-Added in the statistical analysis page 23 that  "The experiments across the manuscript were repeated at least three times"

2-Included in material and methods (p19) the MW standards used (Precision plus protein Dual color standards) and have also included the MW ladder to Figure 3 western blot.

Hope these modifications clarify your concerns

Thanks once again

Plinio

Reviewer 2 Report

This work characterized the Urine Stem Cell-Derived Extracellular vesicles, and found that the USC-derived EVs are a heterogeneous population.
Functionally, the Evs inhibited T cell function but induced B cell proliferation and IgM antibody secretion. Analysis of the EV contents revealed the presence of BAFF, APRIL, IL-6, and CD40L, all known to play a central role in B cell stimulation. 

The data are clear and interesting, only a bit odd that T cells are suppressed but B cells were activated by these Evs. Although they discussed a group of factors to be potent B cell growth and survival factors such as including IL-6, BAFF, APRIL, and CD40L, but these factors cannot be the factors inhibiting T cells, a bit more detailed characterization to demonstrate the common/different factors in inhibiting T cells and/or activating B cells would improve the quality of the study substantially.

Author Response

Dear Reviewer, 

We would like to thank you for reviewing our work and the constructive comment.

We agree that it is important to address the possible mechanism behind the effect on T and B cells discrepancy. In this regards, we have included a paragraph in discussion (p14) providing addition arguments that could support our observation. 

"The different effects of the USC-derived vesicles on T and B cells, suppressing the former and stimulating the latter, is intriguing, in terms of its biological function, however, the diversity of functions of the molecules they carry could support the finding. For example, IL-10 might be contributing to both effects, given its well know inhibitory effect on T cells functions (73-75) and stimulatory effects on B cells, contributing to their proliferation, differentiation and antibody secretion (73). Besides, the presence of TGFβ in these vesicles, previously reported (26), could be contributing to the inhibitory effect on T cell functions (74), whilst positively modulating B cells functions (75). In addition, apart from the regulatory properties of the miRNA described above on T cells, miRNAs are also able to engage TLR7 (76-78), a main player in B cell activation (79, 80).  Yet, further studies might elucidate other mechanisms behind our observation."

We believe the inclusion of this additional comments has enriched the discussion and has added new insights into the possible mechanisms behind our observation.

Thanks again

Plinio